# Exogenous Application of dsRNA for Protection against Tomato Leaf Curl New Delhi Virus

**DOI:** 10.3390/v16030436

**Published:** 2024-03-12

**Authors:** Fulco Frascati, Silvia Rotunno, Gian Paolo Accotto, Emanuela Noris, Anna Maria Vaira, Laura Miozzi

**Affiliations:** Institute for Sustainable Plant Protection, National Research Council, Strada delle Cacce 73, 10135 Torino, Italysilvia.rotunno@ipsp.cnr.it (S.R.); gianpaolo.accotto@ipsp.cnr.it (G.P.A.); emanuela.noris@ipsp.cnr.it (E.N.)

**Keywords:** RNAi-based vaccination, dsRNA-mediated virus resistance, double-stranded RNAs, geminivirus, DNA viruses

## Abstract

Tomato leaf curl New Delhi virus (ToLCNDV) is an emerging plant pathogen, fast spreading in Asian and Mediterranean regions, and is considered the most harmful geminivirus of cucurbits in the Mediterranean. ToLCNDV infects several plant and crop species from a range of families, including Solanaceae, Cucurbitaceae, Fabaceae, Malvaceae and Euphorbiaceae. Up to now, protection from ToLCNDV infection has been achieved mainly by RNAi-mediated transgenic resistance, and non-transgenic fast-developing approaches are an urgent need. Plant protection by the delivery of dsRNAs homologous to a pathogen target sequence is an RNA interference-based biotechnological approach that avoids cultivating transgenic plants and has been already shown effective against RNA viruses and viroids. However, the efficacy of this approach against DNA viruses, particularly *Geminiviridae* family, is still under study. Here, the protection induced by exogenous application of a chimeric dsRNA targeting all the coding regions of the ToLCNDV DNA-A was evaluated in zucchini, an important crop strongly affected by this virus. A reduction in the number of infected plants and a delay in symptoms appearance, associated with a tendency of reduction in the viral titer, was observed in the plants treated with the chimeric dsRNA, indicating that the treatment is effective against geminiviruses but requires further optimization. Limits of RNAi-based vaccinations against geminiviruses and possible causes are discussed.

## 1. Introduction

Cucurbits are among the most important horticultural vegetables cultivated worldwide. They can be infected by several viral pathogens [1,2,3,4], with considerable yield losses. Among the most damaging viral pathogens of cucurbit crops, tomato leaf curl New Delhi virus (ToLCNDV, genus *Begomovirus*, family *Geminiviridae*) has recently emerged. Originally distributed across the Indian subcontinent, East and Southeast Asia (https://gd.eppo.int/, accessed on 28 February 2024), it spread rapidly in the Mediterranean basin [5,6,7,8,9]. ToLCNDV genome comprises two circular single-stranded (ss)DNA segments referred to as DNA-A and DNA-B. DNA-A harbors six ORFs encoding proteins for replication (AC1-Rep, AC3-REn, AC2-TrAP), symptoms expression (AC4), encapsidation (AV1-CP) and silencing suppressor (AV2-pre-coat), while DNA-B encodes a nuclear shuttle protein (BV1-NSP) and a movement protein (BC1-MP) [10,11,12,13].

Current approaches for the management of viral pathogens are based on the use of chemicals to control vector populations, the development of virus-resistant/tolerant crops, and the integration of different crop management strategies. However, the urgent need to reduce the use of chemical pesticides, as underlined by the 2030 Agenda for Sustainable Development (https://sdgs.un.org/goals, accessed on 28 February 2024), and the continuous emergence of viral diseases, favored by globalization and climate change, necessitates new sustainable pest and disease control strategies [14]. RNAi vaccination is a promising and sustainable biotechnological approach that has been receiving great interest in recent years [15]. It is based on the exogenous application of double-stranded RNAs (dsRNAs) or small RNAs (sRNAs) molecules homologous to the target pathogen and relies on the natural phenomenon of RNA interference (RNAi), an evolutionary conserved RNA-mediated regulatory mechanism, consisting in the sequence-specific degradation of target RNA guided by complementary sRNA molecules [16]. Besides its key role in regulating growth and development of the organisms, RNAi is involved in host defense against pathogens [17]. RNAi is triggered by dsRNA or hairpin RNAs (hpRNA) which are cut by specific nucleases called DICER or DICER-LIKE (DCL) into sRNAs. Then, sRNAs are incorporated into an RNA-induced silencing complex (RISC) and guide the cleavage of complementary ssRNAs, such as messenger RNAs or viral genomic/antigenomic RNAs [17]. Several studies have shown that the exogenous application of dsRNA molecules homologous to viral sequences can induce RNAi-mediated defense in plants [15,18,19]. This approach has been proposed as a valid non-transgenic strategy to control diseases induced by RNA viruses. However, few and contrasting results have been obtained against DNA viruses belonging to the *Geminiviridae* family. A reduction in the disease incidence ranging from 40 to 60% was obtained in plants infected by tomato leaf curl virus or chili leaf curl virus and treated with dsRNAs simultaneously targeting different ORFs [20,21]. Partial symptom remission and reduction in the viral titer was observed in blackgram plants treated with dsRNAs targeting the CP and Rep coding regions of mungbean yellow mosaic virus [22]. However, no detectable antiviral effect mediated by topical application of dsRNAs has been observed until now against tomato leaf curl New Delhi virus in cucurbits [23] or tomato severe rugose virus in tomato [23,24].

In this work, we designed a chimeric dsRNA simultaneously targeting all the ORFs present in the DNA-A of ToLCNDV. We then applied it to zucchini plants and evaluated its ability to elicit an RNAi-mediated antiviral response and eventually to protect zucchini plants from the infection. For this purpose, the chimeric dsRNA was applied on the leaves and systemic symptom development and viral titer were evaluated. In addition, the fate of dsRNAs in healthy zucchini plants was studied.

## 2. Materials and Methods

### 2.1. Biological Material

Zucchini (*Cucurbita pepo* L.) plants, cv. Genovese, were maintained in a growth chamber at 24 °C with a light/dark cycle of 16/8 h. Plants with two fully expanded leaves were used for the bioassays. Plants were inoculated with a ToLCNDV isolated from zucchini (isolate PLAVIT-Z364 (zu271016)) from PLAVIT collection, IPSP-CNR, Torino, Italy; GenBank Id OQ970062 and OQ970061).

### 2.2. Design and Synthesis of dsRNAs

A chimeric dsRNA targeting all the ORFs present in the DNA-A of ToLCNDV (dsRNA-V) was designed, based on GenBank OQ970062 sequence. A dsRNA (761 nt) targeting the nucleocapsid (N) coding region of the tospovirus tomato spotted wilt virus (TSWV) described in [25] was used as negative control (dsRNA-C). Indeed, due to the lack of homology with the ToLCNDV genome, this dsRNA was not expected to show any protective effect. The synthesis of dsRNAs was externalized to the AgroRNA division of the Genolution Company (Gangseo-gu, Seoul, Republic of Korea; https://agrorna.genolution.co.kr/, accessed on 28 February 2024).

### 2.3. Virus Inoculation and Exogenous Application of dsRNAs

Zucchini plants were mechanically inoculated using leaf sap obtained from ToLCNDV-infected plants, diluted in COMAV buffer [26]. In detail, 0.8 g of fresh and symptomatic leaf tissue were homogenized in 8 mL of 2X COMAV buffer and an equal volume of nuclease-free water was added to the mixture. A 50 μL dose was applied on the upper surface of two leaves previously dusted with carborundum. When required, 100 μg of dsRNA were added to the virus inoculum, just before inoculation. The buffer alone was used for negative control plants. In each experiment, three to five plants per treatment were used and the experiment was repeated three times. Symptoms were evaluated at 10 and 30 days post inoculation (dpi) using the following disease severity index (DSI): 0: no symptoms; 1: mild symptoms (leaf mosaic, leaf yellowing); 2: severe symptoms (leaf curling and deformation, leaf mosaic, leaf yellowing, vein thickening, internode shortening and stunting). The presence of viral DNA was checked by a tissue print assay [27], using the last newly emerged leaf that did not receive the inoculum. Statistical significance in the DSI and infection percentage was evaluated by using the non-parametric pairwise Mann–Whitney test (significant *p*-value < 0.05).

### 2.4. Movement and Stability of dsRNAs in Planta

To evaluate the persistence/movement of dsRNAs in healthy zucchini plants, 100 μg of dsRNA-V was mixed with 100 μL of nuclease-free water and mechanically inoculated by gentle rubbing with carborundum on the first fully expanded leaf. Two leaf disks were collected from each treated and first expanded leaf from the apex at 0, 1, 4, 7 and 11 dpi, using three plants for each time point. Just before sampling, leaves were washed with Triton X-100 (0.05%) and water to eliminate residual dsRNAs present on the leaf surface.

Total RNA was extracted by Spectrum™ Plant Total RNA Kit (Sigma Aldrich, St. Louis, MO, USA). cDNA synthesis was performed with random primers using the High-Capacity cDNA Reverse Transcription Kit (Thermo Fisher Scientific, Waltham, MA, USA), according to manufacturer’s instructions. The presence of dsRNAs was evaluated by end-point PCR using specific primers dsND_endPCR_fw/dsND_endPCR_rv (Appendix A). PCR was performed in a final volume of 25 μL, containing 2.5 μL reaction buffer 10X, 1 μL cDNA template, 200 μM each dNTPs, 0.4 μM each primer, 2 mM MgCl_2_ and 0.2 μL Platinum Taq DNA polymerase (Thermo Fisher Scientific, Waltham, MA, USA). After an initial denaturation step for 3 min at 95 °C, 35 cycles consisting each of 30 s at 94 °C, 30 s at 56 °C and 45 s at 72 °C were performed, followed by a final extension step for 7 min at 72 °C.

### 2.5. Quantitative PCR for Virus and dsRNA Quantification

Quantitative PCR (qPCR) was carried out using iCycler iQ Real-Time PCR Detection System (BioRad Laboratories, Hercules, CA, USA), with the following cycling parameters: 1 cycle at 95 °C for 5 min, 45 cycles, each consisting of 15 s at 95 °C and 1 min at 60 °C. A melting curve was recorded at the end of each run to assess the specificity of amplification. All reactions were performed with three technical replicates. qPCR efficiency was calculated using standard curves constructed with serial dilutions of DNA extracted from infected plants. Data acquisition and analysis were handled by the BioRad iCycler software (version 3.06070) that calculates Ct values and standard curves. Specific primers dsND_qPCR_fw/dsND_qPCR_rv were used to amplify exogenous dsRNAs, primers qToLCNDV_B_fw/qToLCNDV_B_rv [28] were used for amplification of ToLCNDV DNA-B, and primers ZuEF-1A_fw/ZuEF-1A_rv were used for the amplification of the zucchini gene HO702383 coding for the Elongation factor-1α (EF-1A), used as reference gene [29] (Appendix A). For each biological replicate, qPCR was performed on three technical replicates. The relative quantification was calculated for each sample using the ΔCt method, where ΔCt is |Ct_target_ − Ct_EF-1A_|. Statistical significance in the dsRNA quantification was evaluated by using the non-parametric Kruskal–Wallis test (significant *p*-value < 0.05).

### 2.6. Small RNA Analysis by High-Throughput Sequencing (HTS)

Small RNA population originating from dsRNA-V was analyzed in samples collected for the dsRNA movement investigation at 4 dpi. Both dsRNA-V-treated and newly emerged non-treated leaves were considered for the analysis. Libraries preparation and sequencing were performed by Novogene Company (Cambridge, UK; https://www.novogene.com/eu-en/, accessed on 28 February 2024). Adapter removal and low-quality filtering were performed with fastp version 0.21.0 [30]; ribosomal removal was performed with bbmap version 38.7 [31]. Clean reads were mapped to the dsRNA-V and to the ToLCNDV DNA-A sequences using bowtie v1-1-2 [32] with 0 mismatches. Mapping results were visualized using ad hoc R scripts. Sequence data have been submitted to the Sequence Read Archive (SRA) with the BioProject ID PRJNA1073428.

## 3. Results

### 3.1. Design and In Vitro Synthesis of the Chimeric ToLCNDV-Targeting Exogenous dsRNA

The DNA-A genome encodes viral proteins involved in processes fundamental for the infection, such as viral replication, encapsidation, silencing suppression and movement; this molecule was therefore selected as target of exogenous dsRNAs. A chimeric dsRNA (dsRNA-V) targeting all the DNA-A ORFs was designed, including 270 bp homologous to the 2133–2402 nt region of ToLCNDV-DNA-A GenBank Id. OQ970062), encompassing the AC1-Rep and AC4 ORFs, encoding proteins involved in replication and symptoms expression. This region was followed by a second 270 bp sequence homologous to the region 245–514 nt within the AV1-CP ORF, fundamental for encapsidation, and AV2 ORF encoding a protein potentially involved in silencing suppression. The last 270 bp region selected covered the region 1186–1455 nt, included in the ORFs encoding the AC2-TrAP and AC3-Ren proteins (Figure 1). Overall, the chimeric dsRNA-V had a length of 810 bp, compatible with the fact that dsRNA fragments of 600–900 nt are the most efficient in protecting plants from viral infection [18].

### 3.2. Protection Efficacy of the Chimeric dsRNA Molecule against ToLCNDV

Following inoculation, we monitored by visual inspection the onset of ToLCNDV symptoms in zucchini plants treated with either the homologous dsRNA-V or the control dsRNA-C, or in those inoculated only with the virus. Typical ToLCNDV symptoms were evaluated according the DSI described in the Materials and Methods section (Figure 2).

Two different time points after virus inoculation were considered. At 10 dpi, 15% of the inoculated plants that did not receive any dsRNAs showed mild viral symptoms, such as leaf mosaic, and yellowing (DSI = 1), while 85% of these plants showed no symptoms (DSI = 0). At the same time point, 38% of the plants treated with the control dsRNA-C were moderately symptomatic (DSI = 1), whereas 62% of them did not show any symptoms (DSI = 0). Finally, at the same time point, 8% of the plants inoculated and treated with the dsRNA-V showed typical mild viral symptoms (DSI = 1), while 92% of them was asymptomatic (DSI = 0). At this time point, severe symptoms (DSI = 2) were not observed in any of the considered treatments. These results highlight a significant higher number of symptomatic plants in the case of dsRNA-C treated plants in respect to dsRNA-V-treated plants; an intermediate situation was observed in the case of the untreated inoculated plants (Figure 3a,b). At 30 dpi, 77% of the untreated inoculated plants were symptomatic and all of them showed severe symptoms (DSI = 2). At the same time point, 92% of the plants inoculated with the virus and treated with the control dsRNA-C were symptomatic; in detail 77% showed severe symptoms (DSI = 2) and 15% mild symptoms (DSI = 1). Finally, at this time point, 69% of the plants inoculated and treated with the dsRNA-V were symptomatic; 46% showed severe viral symptoms (DSI = 2); and 23% showed mild symptoms (DSI = 1). A total of 23% of the untreated but inoculated plants, 8% of the inoculated plants treated with dsRNA-C and 31% of those inoculated and treated with dsRNA-V were asymptomatic (DSI = 0) (Figure 3a,b). These results highlight that the difference in the number of symptomatic plants remains even at 30 dpi. Interestingly, a lower number of plants that develop severe symptoms is registered in the case of dsRNA-V-treated plants in respect to dsRNA-C treated or untreated plants, suggesting that dsRNA-V could limit the onset of severe symptoms.

To confirm such symptom evaluation, a tissue print molecular hybridization assay was carried out, at both time points after ToLCNDV inoculation. This molecular assay showed that at 10 dpi, 53% of the plants receiving no dsRNAs and 67% of those treated with dsRNA-C resulted in being infected, while only 24% of plants treated with the dsRNA-V were positive. At 30 dpi, 80% of the untreated plants and 93% of those treated with dsRNA-C were positive, while only 69% of those treated with dsRNA-V showed a hybridization signal (Figure 3c). The difference among these values was statistically significant at both time points, indicating that the exogenous application of dsRNA-V delays not only symptoms appearance but also reduces the percentage of infected plants, thought slightly. It is noteworthy that these results also showed that the treatment with the unrelated dsRNA-C, selected as negative control, slightly favored ToLCNDV infection, suggesting that it could be not so neutral as predicted in respect to the infection process of ToLCNDV in this host.

To further investigate the effect of exogenous dsRNAs on ToLCNDV infection, we measured the ToLCNDV titer in plants by qPCR. As shown in Figure 4a, at 10 dpi, the average quantity of viral DNA present in plants treated with dsRNA-V was similar to untreated plants, while the application of dsRNA-C apparently increased ToLCNDV accumulation. When plants were analyzed at 30 dpi, a slight reduction in the ToLCNDV titer was observed in dsRNA-V-treated plants in respect to dsRNA-C treated and untreated plants, even if an overall increase in the ToLCNDV titer, due to the progression of the infection, was recorded in all plants (Figure 4a).

To better clarify such results, we decided to quantify by qPCR the amount of viral DNA considering only the plants confirmed infected by molecular assay at 30 dpi (end of the experiment). As shown in Figure 4b, this analysis confirmed the increased viral accumulation from 10 dpi to 30 dpi; no difference in viral titer could be observed among plants subjected to the different treatments, at either of the two time points. The difference in viral titer observed at 10 dpi mainly results from the plants that did not develop the infection. This result indicates that the RNAi-based defense activated by the dsRNA-V effectively represses ToLCNDV multiplication at the onset of the infection process, thus preventing the establishment of a systemic infection. However, once the infection has started, the application of dsRNA-V has no curative effect.

### 3.3. Systemic Movement of ToLCNDV-Targeting Exogenous dsRNA in Zucchini

To explain the results obtained in the challenge experiments, we investigated the persistence and systemic movement of the dsRNA-V in zucchini plants, in the absence of virus infection at 1 h post treatment and at 1, 4, 7 and 11 days post treatment (dpt). For this, end-point PCR (ePCR) and quantitative real-time PCR (qPCR) were carried out in leaves treated with the dsRNA (local persistence) and in young newly emerged leaves (systemic movement). Although the quantity of dsRNA-V decreased slightly since its application on leaves, it could be detected until 11 dpt, indicating a relatively long-lasting persistence (Figure 5a,b). Interestingly, dsRNA molecules could be detected also in untreated newly emerged leaves already 1 h after the application, though with variable results among plants. In these leaves, the dsRNA-V could be detected until 11 dpt (i.e., at the end of the experiment), with the maximum observed at 4 dpt (Figure 5a,b). However, the amount of dsRNA measured systemically moving in the newly emerged leaves was 10^−4^–10^−5^ times lower compared to treated leaves at the same time point, indicating that only a reduced percentage of the dsRNA moves systemically in zucchini plants. The lack of a reduction in dsRNA quantity, observed in the untreated leaves at the end of the experiment, may be due to the abundance of dsRNA molecules applied in the treated leaves that maintains a continuous movement of dsRNAs from treated to newly emerged untreated leaves and suggests that the quantity of dsRNA molecules used in the experiments is not a limiting factor.

### 3.4. Analysis of the sRNAs Population Originating from Exogenous dsRNA-V

To investigate if the exogenous dsRNA-V was processed by Dicer enzymes and was able to originate silencing effector molecules (i.e., siRNAs), we sequenced the small RNAs (sRNAs) population present in treated and newly emerged non-treated zucchini leaves, 4 days post exogenous dsRNA application. After adapter and ribosomal removal and quality filtering, 72 to 46 million of clean reads were obtained (Appendix A). Sequences with length ranging from 18 to 29 nt were selected for further analyses. In the leaves treated with exogenous dsRNAs, 3.56% of the total number of reads mapped to the dsRNA-V and corresponded to all the three regions of the viral DNA-A previously selected as the target (Appendix A and Figure 6a). When newly emerged untreated leaves were considered, a limited number of reads, corresponding to the 0.0007% of the clean reads, mapped to the corresponding viral regions (Appendix A and Figure 6a). In both cases, the presence of hot and cold spots but no bias in strand polarity was observed (Figure 6a). These results show that the dsRNA-V is effectively introduced and processed into the plant, thus originating sRNAs that could guide the cleavage of cognate sequences. However, in line with the limited amount of dsRNAs that move systemically into the plant, only few dsRNA-derived sRNAs were found in the newly emerged untreated leaves.

As expected, in both treated and newly emerged untreated leaves, the population of plant-derived sRNAs was enriched in 21 nt-long sRNAs; when non-redundant sRNAs were considered, an enrichment of 24 nt-long sRNAs was observed (Appendix A). Focusing on the sRNAs specifically originating from the exogenous dsRNA-V in treated leaves, we observed a basically uniform length distribution and no well-defined peaks related to a specific length were found (Figure 6b). However, considering dsRNA-derived sRNAs in the newly emerged untreated leaves, a small peak of 24 nt-long sRNAs is observed. The peak at 24 nt in length become more evident when non-redundant reads are considered in both treated and newly emerged untreated leaves (Figure 6b). The absence of a peak at 21 nt and 22 nt may suggest that the exogenously applied dsRNAs were not canonically processed into sRNAs and could contribute to explain the reduced level of protection observed in dsRNA-V-treated plants.

## 4. Discussion

Previous works performed using RNAi-mediated transgenic resistance indicated that this strategy could be successfully employed to protect plants against ToLCNDV infection [33,34,35]. Transgenic tomato plants expressing artificial microRNAs targeting the overlapping region of the AV1 and AV2 proteins [33] or the AC1-Rep coding region [34] were tolerant to virus infection. Also, plants producing artificial trans-acting small interfering RNAs simultaneously targeting the viral transcripts coding for the AC3-REn and transcriptional activator protein AC2-TRaP did not show viral symptoms and accumulated low amounts of viral DNA [35]. However, the concerns about potential negative effects of genetically modified plants on human health and environment, together with their limited acceptance by consumers, requires the development of alternative non-transgenic strategies. Among them, those exploiting RNAi, such as exogenous application of dsRNAs, seems to have a higher chance to be accepted by the consumers [36].

In this study, we investigated the effect of the topical application of dsRNA molecules against ToLCNDV, an economically important geminivirus, contributing to shed light on the effectiveness of this approach against geminiviruses. The results showed that the topical application of a chimeric dsRNA targeting different regions of the DNA-A can limit the onset of severe symptoms and reduce the viral infection rate. These findings indicate that targeting simultaneously different ORFs coding for proteins involved in different steps of the viral life cycle may be a winning strategy, particularly for geminiviruses. Our results differ from those obtained by [23] where the efficacy of two different dsRNAs targeting the AV1-CP and BC1-MP coding regions was assayed in cucurbits. In this case, no protection against ToLCNDV infection was reported. Nonetheless, our results are in line with those obtained using chimeric dsRNAs targeting different viral ORFs in the tomato-tomato leaf curl virus pathosystem [20], thus underlining the importance of the selection of the viral target region [25].

Although the dsRNA-based vaccination approach appears able to protect plants from ToLCNDV infection, the level of protection obtained is quite low and effective only at the beginning of the infection process. This could be related to the life cycle of geminiviruses, or DNA viruses in general. In the RNA-directed silencing of DNA viruses, dsRNAs originated by the base-pairing of overlapping transcripts are initially recruited into the core post-transcriptional RNAi pathway as in the case of RNA viruses; then, if the RNA silencing is weak and viral RNAs continue to accumulate, there could be a transition to RNA-dependent DNA methylation (RdDM) with the consequent transcriptional inhibition [37]. However, externally administered dsRNA fragments have been recently observed to be processed by a non-canonical biogenesis pathway that produce a ladder of sRNAs of 18–30 nt in length, instead of the canonical 21 and 22 nt sRNA species (this study, [25,38]). These not canonical sRNAs seems to be not able to induce transitive amplification [38]. Therefore, it could be hypothesized that sRNAs originated from exogenous dsRNAs would be not efficient in activating the RNAi machinery and/or leading to the RdDM transition. In this case, only a limited protection against viral infection would be guaranteed. The use of selected exogenous ds-sRNAs targeting the viral sRNAs hot spots could be evaluated as a possible more efficient alternative, as highlighted in the pathosystem *N. benthamiana*/African cassava mosaic virus [39]. Indeed, it seems that the vast majority of sRNAs generated by external dsRNAs have non-canonical AGO-incompatible sizes and that only a small fraction of external dsRNAs is processed by DCL2 and DCL4 into AGO2 and AGO1-compatible siRNA duplexes [38,40]. In this context, the ds-sRNAs-based strategy would bypass the non-canonical synthetic pathway and would prevent the overloading of core components of the RNAi machinery with ineffective sRNAs, possibly improving the protective effect. Further in-depth investigations on the molecular mechanisms underpinning the processing and functions of externally applied dsRNAs are needed in order to set up effective RNAi-based protection strategies, particularly against DNA viruses.

The difference between DNA and RNA viruses is evident also when the curative efficacy of the dsRNA treatment is evaluated on plants which are already infected. No protective effect was observed in zucchini plants inoculated with the DNA virus ToLCNDV three days after treatment with dsRNAs targeting the coat protein (AV1-CP) or the movement protein (BC1-MP) coding regions [23]. This result is in contrast with those reported for zucchini plants challenged with the RNA virus cucumber green mottle mosaic virus (family *Virgaviridae*), for which the dsRNA treatment reduced symptom severity and limited disease progression [23].

A certain level of protection against geminiviruses was obtained until now with dsRNAs, mainly when they were applied together with the viral inoculum (this study; [20,39,41]); no protection was observed when the virus is delivered through viruliferous whiteflies on plants treated with dsRNAs 24 h before [24]. The contrasting results could be partially related to the poor systemic translocation and movement of the naked dsRNAs in plants. Indeed, the persistence and movement of the exogenously applied dsRNAs in the host plant is crucial. In this respect, we observed that exogenous dsRNAs persist on zucchini leaves for several days (11 dpi), but, even if the quantity of dsRNA molecules provided to the plants is high, only a small amount move systemically; indeed, this translocation occurs within the plant shortly after treatment and is stably maintained. These findings are in line with those previously reported in *N. benthamiana* [25], tobacco [42] and cucurbits [43] and prompts the need of developing ad hoc nanocarriers able to facilitate the delivery and increase the stability of dsRNAs in the plant [44] and the dsRNA-delivered siRNA production [45].

It is noteworthy that the synergistic effect induced by the dsRNA-C targeting TSWV on ToLCNDV infection could be explained by the generation of sRNAs from the dsRNA-C, possibly interfering with the host transcriptional or post-transcriptional response of the host. Indeed, several evidence indicated that virus-derived small interfering RNAs (vsiRNAs) may perturb the host responses to viral infection and attenuate their defense system [46,47,48,49,50]. The analysis of TSWV-derived siRNAs in infected tomato led to identify more than 2000 vsiRNAs potentially able to interact with the host transcriptome, suggesting that vsiRNAs may interfere with host–virus interaction [51]. Further investigations are needed to clarify this aspect.

To conclude, we showed that the RNAi-based vaccination may have a protective effect also against ToLCNDV. Moreover, we pointed out that the use of chimeric dsRNAs simultaneously targeting different viral coding regions is an effective strategy, thus contributing to extend the use of this technology to control economically important viruses belonging to the *Geminiviridae* family. However, we highlighted that further studies aiming to investigate the processing of exogenous dsRNAs in plant and consequently improve the efficacy of exogenously applied dsRNAs for plant virus control are needed, in order to make the RNAi-based vaccination a real eco-friendly alternative for crop protection.

## Figures and Tables

**Figure 1 viruses-16-00436-f001:**
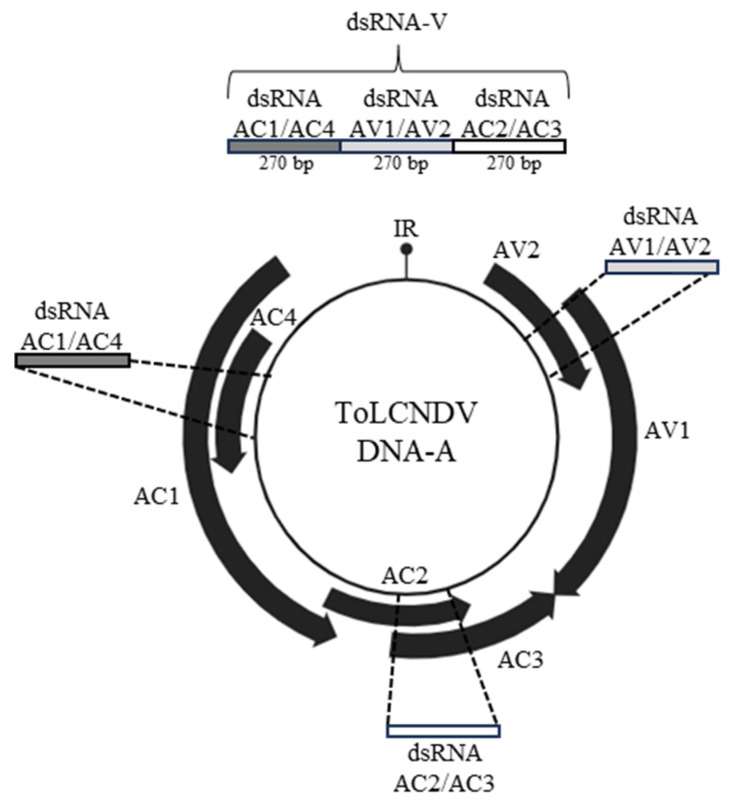
Schematic representation of the chimeric dsRNA (dsRNA-V) used in this study. Figure created with BioRender.com.

**Figure 2 viruses-16-00436-f002:**
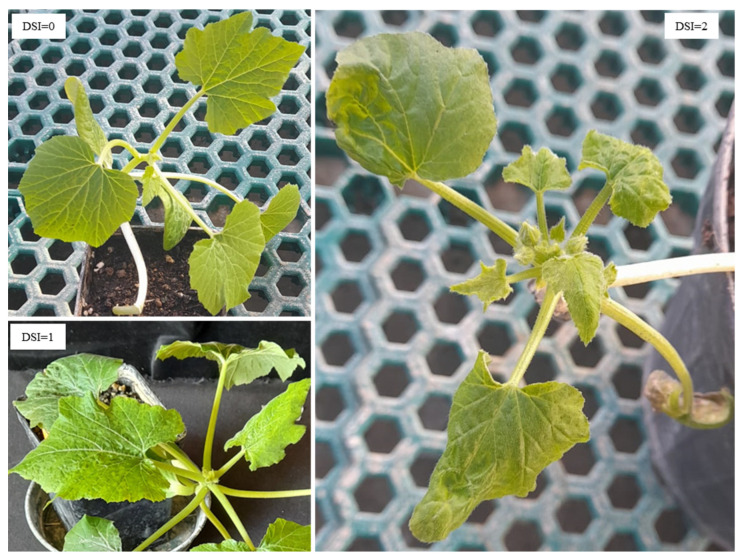
Examples of ToLCNDV symptoms according to the disease severity index (DSI) evaluation; DSI = 0, no symptoms; DSI = 1, leaf mosaic, leaf yellowing; DSI = 2, leaf curling and deformation, leaf mosaic, leaf yellowing, vein thickening, internode shortening and stunting.

**Figure 3 viruses-16-00436-f003:**
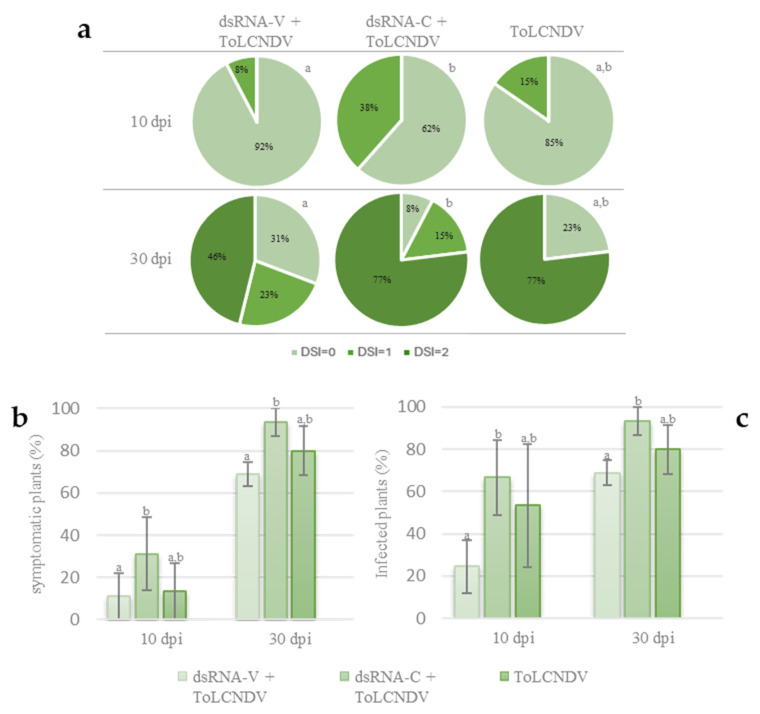
Efficacy of the dsRNA treatment against ToLCNDV on zucchini plants. (**a**) Graphic representation of the evaluation of symptoms caused by ToLCNDV in zucchini plants based on a disease severity index (DSI) ranging between 0 (no symptoms) and 2 (severe symptoms); different letters near the pie charts indicate statistically significant differences; (**b**) percentage of symptomatic plants according to visual symptoms evaluation; (**c**) percentage of infected plants according to molecular hybridization assay. Vertical lines on each bar represent standard errors. Different letters indicate statistically significant differences (*p* < 0.05, Mann–Whitney test).

**Figure 4 viruses-16-00436-f004:**
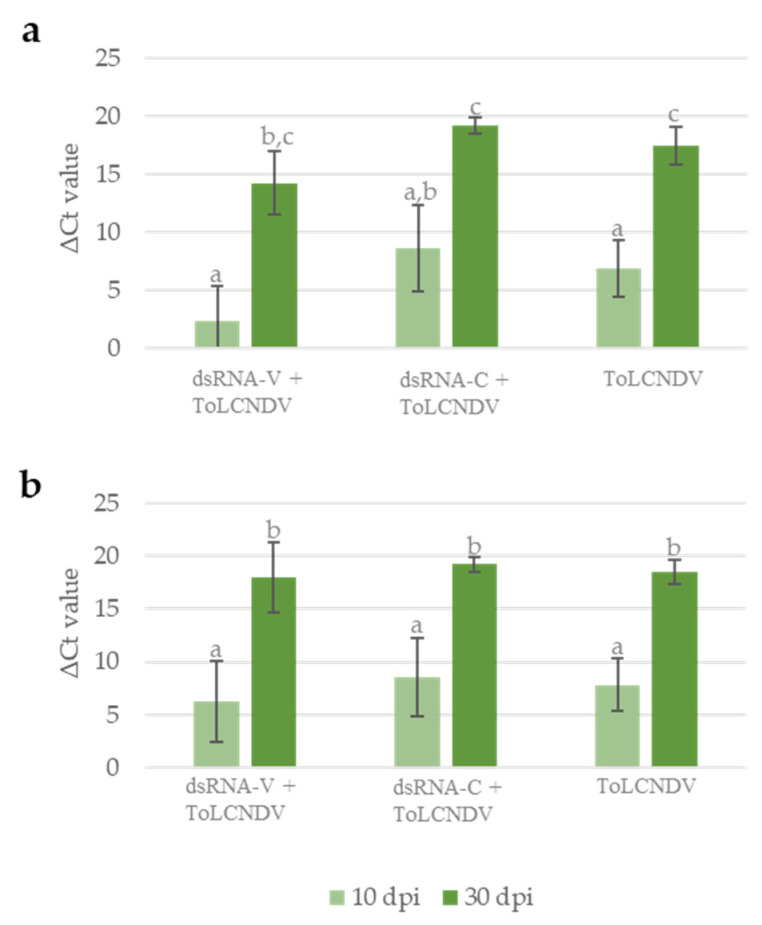
ToLCNDV titer evaluated as the ΔCt value between the Ct value measured for the ToLCDNV-DNA-B and the Ct value measured for the reference gene EF-1A, considering (**a**) all the plants inoculated with ToLCNDV or (**b**) only the plants confirmed infected by molecular hybridization at 30 dpi (end of the experiments). Vertical lines on each bar represent standard errors. Different letters indicate statistically significant differences (*p* < 0.05, Kruskal–Wallis test).

**Figure 5 viruses-16-00436-f005:**
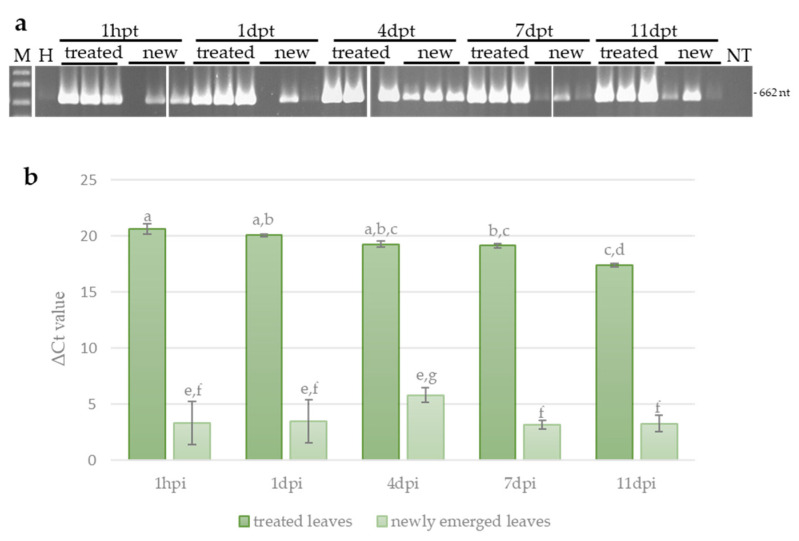
Persistence and systemic movement of dsRNA-V. (**a**) Detection of dsRNAs-V in treated leaves and untreated newly emerged leaves of zucchini at different time points by end-point PCR. NC, negative control (non-treated plants); NT, no template; hpt, hours post treatment; dpt, days post treatment; M, 1 kb plus DNA Ladder (New England Biolabs, MA, United States). Sizes of amplified fragments are shown on the side. (**b**) Quantification of dsRNA-V by qPCR. The Ct values obtained for the dsRNA were normalized with the Ct values obtained for the EF-1A transcript used as reference. Vertical lines on each bar represent standard errors. Different letters indicate statistically significant differences (*p* < 0.05, Kruskal–Wallis test).

**Figure 6 viruses-16-00436-f006:**
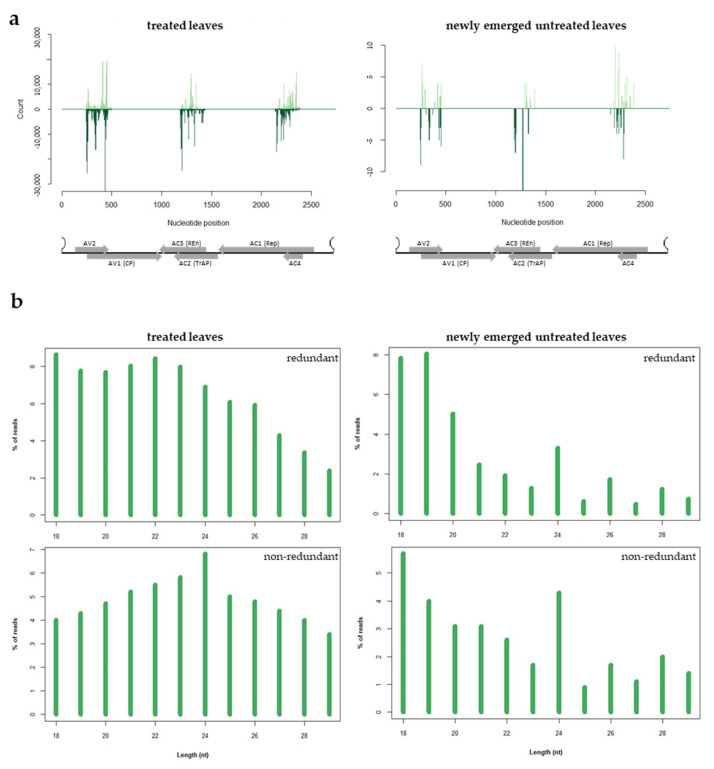
HTS analysis of sRNAs in dsRNA-treated (left panel) and newly emerged untreated (right panel) zucchini leaves, 4 days post treatment. (**a**) Distribution of dsRNA-V-derived sRNAs along the ToLCNDV DNA-A; sense and antisense sRNAs are reported in light and dark green, respectively; ORFs are represented along the viral genome; (**b**) length distribution of the sRNAs reads derived from the dsRNA-V.

## Data Availability

HTS data are available at the Sequence Read Archive (SRA) with the BioProject ID PRJNA1073428.

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
