# Peer review of "Exogenous Application of dsRNA for Protection against Tomato Leaf Curl New Delhi Virus"

_viruses, 2024, doi:10.3390/v16030436_

Round 1
Reviewer 1 Report
Comments and Suggestions for Authors
Dear Authors see my comments below.
Critical comments on the manuscript titled "Exogenous Application of dsRNA for Protection against Tomato Leaf Curl New Delhi Virus":
Comment 1: Is there any variation in symptom severity among the dsRNA-V, dsRNA-C, and untreated control plants after virus infection is confirmed? The manuscript should include this information. Since the percent disease incidence data only indicates the presence or absence of the virus, including severity data is essential. If exogenously applied dsRNA effectively limits disease severity without significantly affecting crop yield, its efficacy can be better demonstrated. Consider using a disease scale to quantitatively measure severity over time.
Comment 2: In the qPCR assay, considering the primary goal is to observe a reduction in virus load, it is suggested to use absolute quantification with a standard curve instead of the delta Ct method (relative quantification). This approach would enable a more accurate comparison of virus load differences between treatments, leading to a more robust conclusion.
Comment 3: Explore potential reasons for the strategy's ineffectiveness, even with the use of chimeric dsRNA containing most viral ORFs and its persistent nature in the host. Consider investigating whether inefficient target sequence selection leads to reduced production of effective siRNAs. Suggest performing tests like stem-loop PCR to confirm the presence of virus-induced siRNAs, referencing Konakalla et al. (2021).
Comment 4: Discuss the observed lack of reduction in the concentration of exogenously applied dsRNA over time, both in local and systemic leaves. Explore possible reasons for this phenomenon to enhance the manuscript's clarity.
Comment 5: Ensure that the statistical analysis used in the experiment is explicitly mentioned in the material section of the manuscript for transparency and reproducibility. This information is crucial for readers to evaluate the robustness of the study's findings.
Author Response
Comment 1: Is there any variation in symptom severity among the dsRNA-V, dsRNA-C, and untreated control plants after virus infection is confirmed? The manuscript should include this information. Since the percent disease incidence data only indicates the presence or absence of the virus, including severity data is essential. If exogenously applied dsRNA effectively limits disease severity without significantly affecting crop yield, its efficacy can be better demonstrated. Consider using a disease scale to quantitatively measure severity over time.
Answer: Data on symptoms severity have been included in the MS according to Reviewer’s advice (see paragraph 3.2 of the Result section and Figure 2 and 3)
Comment 2: In the qPCR assay, considering the primary goal is to observe a reduction in virus load, it is suggested to use absolute quantification with a standard curve instead of the delta Ct method (relative quantification). This approach would enable a more accurate comparison of virus load differences between treatments, leading to a more robust conclusion.
Answer: The relative quantification methods allow to compare the amount of virus in biological samples such as different infected plants or plant tissue samples. The absolute quantification protocol allows to determine, as a final result, the number of viral genomic copies (genomic units, GU) present in an experimental sample and ultimately to refer this value to the number of plant GUs (Noris and Miozzi, 2015, Real-Time PCR Protocols for the Quantification of the Begomovirus Tomato Yellow Leaf Curl Sardinia Virus in Tomato Plants and in Its Insect Vector). Since the target of the qPCR assay in this MS was to investigate the effect of the treatments on the changes in viral load, in comparison to untreated plants, we choose to use a relative quantification approach. This choice was in line with previous analogous studies (Maffei et al., 2014, DOI 10.1007/s00572-013-0527-6; Miozzi et al., 2011, doi:10.1094 /MPMI -05-11-0116; Wu et al. 2014 DOI: 10.1371/journal.pone.0097038; Jarosová et al. 2010. DOI: 10.1186/1471-2229-10-146). It is worth noting that Jarosová et al. 2010 (DOI: 10.1186/1471-2229-10-146) didn’t observe any significant differences between the absolute and relative quantifications of viral titre in plant.
Comment 3: Explore potential reasons for the strategy's ineffectiveness, even with the use of chimeric dsRNA containing most viral ORFs and its persistent nature in the host. Consider investigating whether inefficient target sequence selection leads to reduced production of effective siRNAs. Suggest performing tests like stem-loop PCR to confirm the presence of virus-induced siRNAs, referencing Konakalla et al. (2021).
Answer: The processing of dsRNA-V and the presence of derived sRNAs have been now investigated through high-throughput sequencing and Results and Discussion sections have been modified accordingly (see paragraph 2.6 in the M&M section, paragraph 3.4 in Results section and Discussion lines 370-371). The raw sequencing data have been submitted to SRA-NCBI and are available at https://dataview.ncbi.nlm.nih.gov/object/PRJNA1073428?reviewer=qp3g3lpe5puuccc7fakcope4lg
Comment 4: Discuss the observed lack of reduction in the concentration of exogenously applied dsRNA over time, both in local and systemic leaves. Explore possible reasons for this phenomenon to enhance the manuscript's clarity.
Answer: According to Reviewer’s indication, we inserted in the MS possible reasons on the observed lack of reduction in the concentration of applied dsRNAs (see paragraph 3.3 in the Results section and the Discussion)
Comment 5: Ensure that the statistical analysis used in the experiment is explicitly mentioned in the material section of the manuscript for transparency and reproducibility. This information is crucial for readers to evaluate the robustness of the study's findings.
Answer: Statistical analysis has been added in the M&M section
Reviewer 2 Report
Comments and Suggestions for Authors
This was an interesting manuscript that needs moderate revision to improve clarity and grammar.
The authors should check the journal guidelines in regards to citing references. For example (line 27) should read as “by a tissue print assay [27].”
Line 68. As previous note: Delgado-Martin is reference 23 as used in line 69.
Line 143 is a duplication of line 34-35.
Many sentences are very long and so can be difficult to follow.
Fig 2.A Difficult to see the differences in the symptoms in these photos.
Fig. 2B. In 10dpi, the standard error bars are overlapping which suggests that the values are not significantly different as the bar graph indicates.
Lines 205-215. I found this paragraph difficult to understand due to the extremely long sentences. Could be improved by revision for clarity.
Fig. 3. Interpretation could be improved if each set of three values were analyzed separately.
Line 237. Use superscript for 104 and 105
Line 252-3. Add reference after ‘infection’.
Lines 258-261. There’s no guarantee that the general public will be willing to accept produce treated with a recombinant molecule.
Line 276 Lifecycle – not lifestyle.
Comments on the Quality of English LanguageThis MS would benefit greatly from a revision for grammar and formatting.
Author Response
Moderate editing of English language required
Answer: MS has been revised for English language
This was an interesting manuscript that needs moderate revision to improve clarity and grammar.
Answer: We reviewed the text in order to improve clarity and grammar
The authors should check the journal guidelines in regards to citing references. For example (line 27) should read as “by a tissue print assay [27].”
Answer:Corrected
Line 68. As previous note: Delgado-Martin is reference 23 as used in line 69.
Answer:Corrected
Line 143 is a duplication of line 34-35.
Answer:Corrected
Many sentences are very long and so can be difficult to follow.
Answer:The text has been revised
Fig 2.A Difficult to see the differences in the symptoms in these photos.
Answer:We have inserted in the MS, as Figure 2, new photos showing examples of symptoms used for DSI evaluation
Fig. 2B. In 10dpi, the standard error bars are overlapping which suggests that the values are not significantly different as the bar graph indicates.
Answer:In general, a gap between bars does not ensure significance, nor does overlap rule it out (Krzywinski, M., Altman, N. Error bars. Nat Methods 10, 921–922 (2013). https://doi.org/10.1038/nmeth.2659”, Nature Collection “Points of Significance”). Whether or not standard errors intervals overlap does not imply the statistical significance of the parameters of interest (“Payton ME, Greenstone MH, Schenker N. Overlapping confidence intervals or standard error intervals: what do they mean in terms of statistical significance? J Insect Sci. 2003;3:34. doi: 10.1093/jis/3.1.34. Epub 2003 Oct 30. PMID: 15841249; PMCID: PMC524673”). In this case, the overlapping of the error bars could be linked to the number of tested samples and the use of a statistical test is recomended to verify the significance of the observed differences. We check the statitical analysis performed as described in the M&M section and we confirm that the values are significantly different.
Lines 205-215. I found this paragraph difficult to understand due to the extremely long sentences. Could be improved by revision for clarity.
Answer:Corrected
Fig. 3. Interpretation could be improved if each set of three values were analyzed separately.
Answer:We modified Fig.3 according to referee advice
Line 237. Use superscript for 104 and 105
Answer:Corrected
Line 252-3. Add reference after ‘infection’.
Answer:Done
Lines 258-261. There’s no guarantee that the general public will be willing to accept produce treated with a recombinant molecule.
Answer:We agree that the acceptance of non-transgenic RNAi-based technologies by the consumers need to be further explored; however, given the degradability of RNA molecules and their low environmental impact, it is plausible that these tecnologies, if well explained to the public, would be more acceptable than transgenic approaches. We have modified the sentence and added a reference that investigated the consumer acceptance of topical RNAi products for plant protection.
Line 276 Lifecycle – not lifestyle.
Answer:Corrected
This MS would benefit greatly from a revision for grammar and formatting.
Answer:The MS has been revised for grammar and formatting
Round 2
Reviewer 1 Report
Comments and Suggestions for Authors
Dear Authors,
Change the references according to the journal guidelines.
Thank you